# IEA: Inner Ensemble Average within a convolutional neural network

## Abstract

Ensemble learning is a method of combining multiple trained models to improve model accuracy. We propose the usage of such methods, specifically ensemble average, inside Convolutional Neural Network (CNN) architectures by replacing the single convolutional layers with Inner Average Ensembles (IEA) of multiple convolutional layers. Empirical results on different benchmarking datasets show that CNN models using IEA outperform those with regular convolutional layers and advances the state of art. A visual and a similarity score analysis of the features generated from IEA explains why it boosts the model performance.

## 1 Introduction

Ensemble learning (Rokach, 2010; Zhou, 2012) is the method of combining multiple models trained over the same dataset or a random set of datasets to improve the model performance. The methods of ensemble have been widely used in deep learning (Qiu et al., 2014; Dietterich, 2000; Drucker et al., 1994) to improve the overall model accuracy. Combining neural networks in ensemble is known to reduce the variance in prediction, in other words, it helps the network to generalize more than the usage of one network (Krogh & Vedelsby, 1995; Geman et al., 1992; Zhou et al., 2002). The work by Stahlberg & Byrne (2017) proposed a method of combining multiple models where it unfolds the ensemble into a larger network. Lee et al. (2015) discussed the power of ensemble in training CNN and proposed a method for training ensemble by a specific loss function rather than averaging the predictions of the models.

Convolutional Neural Networks (CNNs) (Lecun et al., 1998) are extremely successful architectures that are widely used in different areas such as computer vision (Krizhevsky et al., 2012), text analysis (dos Santos & Gatti, 2014; Lai et al., 2015), and even general temporal sequence problems (Bai et al., 2018). CNN is a biologically inspired simulation of the cats visual cortex (Hubel & Wiesel, 1968). Usually, CNN is composed of a convolutional layer followed by a pooling layer. Alternating convolutional layers and pooling layers are stacked to introduce more depth to the model being constructed. We call the connection of convolutional layers and pool layers a features extraction head. The features extraction head may or may not be connected to a fully-connected layer based on the application of the deep model. One important feature of convolutional layers that the weights are shared for creating the features. By having this feature of shared weights, convolutional layers do not strongly contribute to the total parameter size of the deep model unlike the contribution of the fully-connected layer.

In this work, we show empirically, visually and through similarity score analysis that replacing ordinary convolutional layer by an Inner Ensemble Average (IEA) of convolutional layers in a CNN can reduce the variance of this model. We propose it as a convenient technique to improve the performance of CNN predictive models.

This work is organized as follows, section 3 defines IEA mathematically. Section 4 shows the experiments performed on different benchmark datasets including MNIST (LeCun, 1998), rotated-MNIST (Larochelle et al., 2007), CIFAR-10, and CIFAR-100 (Krizhevsky & Hinton, 2009) using well-known deep CNN architectures. We show the results of convolutional-layer-only models, IEA of convolutional layers models and ensemble of both techniques. In section 5, a visual analysis of the features generated by IEA is performed. Section 5 also introduces a metric to measure the similarity between the features generated from IEA and convolutional layers.

## 2 RELATED WORK

Ensemble is one of the meta-algorithms that combines a set of independently trained networks into one predictive model in order to improve performance. Other methods include bagging and boosting that reduce predictive variance and bias (Opitz & Maclin, 1999). Ensemble by averaging several models is one of the most common approaches and extensively adopted in modern deep learning practice, especially in related contests (Russakovsky et al., 2015; Rajpurkar et al., 2016). One variation is to ensemble the same model at different training epochs (Qiu et al., 2014). Though inspired by ensemble methods that average independent networks, the proposed IEA structure is fundamentally different in that a) it replicates small units, namely convolutional layers, within CNN structures and b) the replications are trained jointly.

Our work is reminiscent of Maxout (Goodfellow et al., 2013), which replicates weight matrices and take the maximum from the features produce by those linear layers. Maxout can be considered as a special activation method that behaves as a learnable piecewise-linear function. IEA differs from Maxout since IEA averages convolutional layers after non-linearity layers (Figure 1). While Maxout is designed to approximate any activation function, our intuition is as follows: an ensemble of models are used essentially to reduce variance in other terms to increase models generalization. We hypothesize that the overall variance of the model can be decomposed into sub-layers within the model and each layer contributes somehow into this variance. If we used inner ensemble average we will reduce the sub-variances of each layer resulting in an overall variance reduction. Therefore, we primarily explore the averaging method in our setting.

ResNet (He et al., 2015) and its variants (Huang et al., 2017; Szegedy et al., 2017) have become standard for recognition tasks. One can argue that the concept of IEA bears some similarity with the additive behaviors of ResNet and variants (He et al., 2016). Wide Resnet (Zagoruyko & Komodakis, 2016) explores the idea that convolutional layers can have larger dimension while reducing the overall model depth. ResNext (Xie et al., 2017) also makes the model wider by introducing complicated pathways within Resnet blocks and achieved competitive results with more shallow networks. IEA further exploits the possibility of increasing block width by simply plugging into existing models, replicating any convolutional layers within IEA. In our paper, Wide ResNet and ResNext with IEA are intensively experimented and analyzed in comparison with regular ones. Some other widely-used CNN architectures are also explored.

## 3 INNER ENSEMBLE AVERAGE (IEA)

The IEA concept can be applied to a CNN architecture by replacing any convolutional layer ($C_{layer}$) with average ensemble of several replications of a convolutional layer (along with the following batch normalization (Ioffe & Szegedy, 2015) and activation layer, which means averaging is performed after the non-linearity). In any deep CNN architecture to use the IEA concept the $C_{layer}$ is replaced by $C_{IEA}$. The $C_{IEA}$ is defined as follows:

$$C_{IEA} = \frac{1}{m}(\sum_{i=1}^{m} C_{layer=i}) \tag{1}$$

where $m = \{x \,|\, x \in \mathbb{N}^+ - \{1\}\}$ is the number of inner convolutional layers which is a design choice. An illustration of IEA concept within a CNN is found in Figure 1.

When using IEA, the same settings of the replaced convolutional layer are applied to each IEA element individually.

## 4 EXPERIMENTS AND ANALYSIS

In the following section, we empirically evaluate the performance of IEA convolutional layers versus ordinary convolutional layer. In all tests, the number of inner ensembles is set to $m = 3$, to speed up computations. We also compare the results of an ensemble of models trained using convolutional layer and an ensemble of models trained using IEA of convolutional layers. The ensemble of IEA of convolutional layers can be described as an outer and inner ensemble in a CNN.

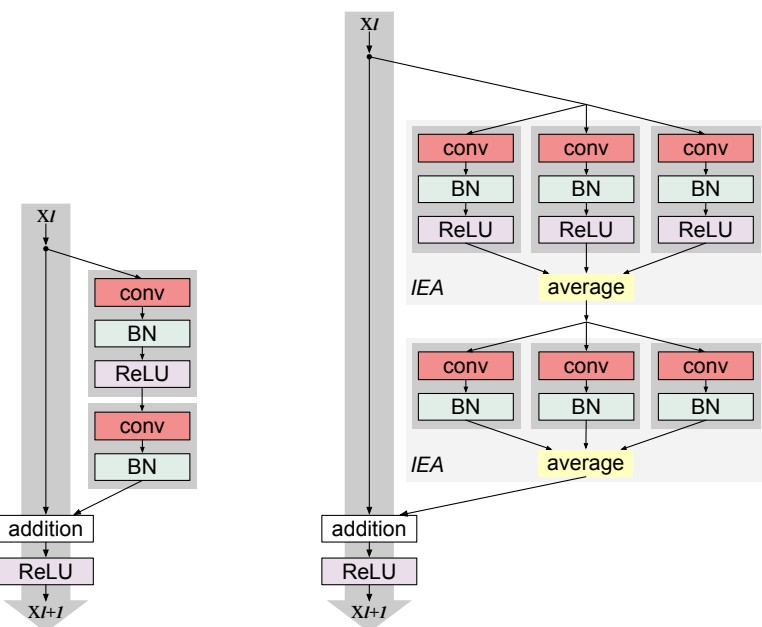

Figure 1: Illustration of IEA structures inside a CNN model. **Left**: a regular ResNet block; **Right**: a ResNet block constructed by replacing each convolutional layer (including the following batch normalization layer and activation layer) with an IEA component.

### 4.1 MNIST AND ROTATED-MNIST DATASETS

The MNIST dataset is one of the earliest datasets in computer vision and machine learning. The MNIST dataset contains 60,000 training set examples and a test set of 10,000 examples of handwritten digits. The rotated-MNIST dataset contains 62000 randomly rotated handwritten digits. The rotated-MNIST dataset is split into a training and test sets of size 50000 and 12000 respectively.

We assembled several vanilla CNN models and trained on both datasets. Each model contains one to three layers. We define a convolutional layer as composition of a weight matrix that applies a 2D convolution over an input signal, the following batch normalization layer and rectified linear unit (ReLU) activation function (Hahnloser et al., 2000). Then a max-pooling layer is stacked after each convolutional layer. Finally, an average-pooling layer is connected before plugging into a classification head. We then expand these small CNN models by replicating any convolutional layers by three and adding an average pooling layer, making it an IEA with $m = 3$. Accordingly, we use $m = 1$ to represent regular convolutional layers. Therefore, we have six different models to test on MNIST/rotated-MNIST datasets.

The models were trained using stochastic gradient descent (SGD) (Bottou, 2010) optimization algorithm with a momentum (Polyak, 1964) set to 0.9 and weight decay (Krogh & Hertz, 1992) set to 5e-4. The initial learning rate is 0.1 and it was divided by 10 for every 100 training epochs. The total number of training epochs is 350. The training datasets were not augmented, and the same initialization settings were kept the same between the models.

Table 1 describes the results on the MNIST test dataset. The usage of IEA of convolutional layers leads to an increase in performance over convolutional-layer-only models. It can be seen that the error rate of the one-layer model decreased by 0.18%. By going deeper with one more layer, the mean error rate improved by 0.76% with just two IEA layers. We achieve a mean error rate of 0.45% on the MNIST dataset by just using three layers deep model.

Even with replications inside the architecture, we can still exploit ensemble of several IEA models to increase the predictive power. Table 2 shows the performance of the average ensemble of three IEA models versus an ensemble of three convolutional-layer-only models. To distinguish from *inner ensemble*, we use $k$ to indicate the number of models in an *outer ensemble* of identical, yet

independently trained networks. In our case, $k$ is 3. As for the one-layer model, both ensemble of IEA of convolutional layers and convolutional-layer-only model show the same performance. The ensemble of IEA models shows a better performance than the ensemble of convolutional layers only based model when two or three layers are used.

Table 1: Test set mean error rates and standard deviation in percentage on the MNIST dataset with different configurations of the tested model architecture. $m = 1$ indicates regular CNNs and $m = 3$ indicates a CNN with IEA that consists of three replications of a convolutional layer.

| Model depth | $m = 1$ | $m = 3$ |
|---|---|---|
| 1 layer | $1.52\pm 0.02$ | $\mathbf{1.34\pm 0.03}$ |
| 2 layers | $1.45\pm 0.03$ | $\mathbf{0.69\pm 0.01}$ |
| 3 layers | $1.46\pm 0.12$ | $\mathbf{0.45\pm 0.02}$ |

Table 2: Test set error rates in percentage of average ensemble models on the MNIST dataset with different configurations of the tested model architecture. $k$ represents the number of individual networks within an ensemble

| Model depth | $m = 1, k = 3$ | $m = 3, k = 3$ |
|---|---|---|
| 1 layer | 1.32 | 1.32 |
| 2 layers | 1.39 | $\mathbf{0.68}$ |
| 3 layers | 1.48 | $\mathbf{0.38}$ |

Table 3 describes the rotated-MNIST test dataset results. In this case, the usage of IEA of convolutional layers still leads to significant improvements over convolutional-layer-only models. It can be seen in the case of the one-layer model that the IEA of convolutional layers model significantly improves the accuracy compared to convolutional-layer-only models. The performance increases when more layers are added (deeper model). We can see a drop of the mean error rate by 0.41% in the case of two layers deep model, and a drop of 0.68% with three layers deep one. These results show the capability of IEA in decreasing the total model variance. Table 4 shows the case of average ensemble of three IEA of convolutional layer models and convolutional-layer-only models. In the case of a one layer deep model, the ensemble of convolutional layer models is better than the ensemble of IEA of convolutional layer models. When more layers are considered, the ensemble of IEA of convolutional layer models outperforms the ensemble of convolutional-layer-only models, except in the case of two layers model were the ensemble of convolutional-layer-only models has a slightly better performance. We attribute this behavior to the selection of hyper-parameters. Also, an error rate of 5.33% is achieved using three layers deep model with IEA of convolutional layers.

Table 3: Test set mean error rates and standard deviation in percentage on the rotated-MNIST dataset with different configurations of the tested model architecture.

| Model depth | $m = 1$ | $m = 3$ |
|---|---|---|
| 1 layer | $47.06\pm 0.33$ | $\mathbf{21.18\pm 0.37}$ |
| 2 layers | $10.47\pm 0.09$ | $\mathbf{10.06\pm 0.05}$ |
| 3 layers | $6.72\pm 0.27$ | $\mathbf{6.04\pm 0.04}$ |

## 4.2 CIFAR-10 DATASET

The CIFAR-10 dataset consists of 60000 color images of size 32x32 and contains 10 classes. There are 50000 training images and 10000 test images. Well-known object detection models including VGG16 (Simonyan & Zisserman, 2014), residual network with 18 layers (ResNet18) and 101 layers (RestNet101) (He et al., 2015), Mobilenet (Howard et al., 2017), Densely Connected Convolutional

Table 4: Test set error rates in percentage of average ensemble models on the rotated-MNIST dataset with different configurations of the tested model architecture.

| Model depth | $m = 1, k = 3$ | $m = 3, k = 3$ |
|---|---|---|
| 1 layer | 45.61 | **19.27** |
| 2 layers | **8.93** | 8.95 |
| 3 layers | 5.86 | **5.33** |

Networks with 121 layers (DenseNet) (Huang et al., 2017), Wide ResNet (Zagoruyko & Komodakis, 2016), and ResNext (Xie et al., 2017) were trained on the CIFAR-10 dataset. Note that Wide ResNet and ResNext are trained on the CIFAR-10 dataset that is augmented using image translation and mirroring. In these models, the convolutional layers were replaced by IEA layers. The training was done using non-augmented training samples. The training configuration is the same as the configurations mentioned in section 4.1. The training and validation curves can be found in the supplementary materials.

Table 5 shows an overall improvement in classification mean error rates on the CIFAR-10 dataset. VGG16 had a 0.64% mean error rate improvement by using IEA of convolutional layers. ResNet18 and ResNet101 were improved by 1.0%, 0.52% mean error rates respectively by using IEA compared to convolutional layer. A significant improvement is seen in MobileNet by 3.09% error rate when using IEA. DenseNet shows an improvement of 0.13% by using IEA of convolutional layers. Wide ResNet and ResNext also improves. To test the effect of models ensemble, Table 6 shows an ensemble of previously mentioned models. We average an ensemble of three models using IEA of convolutional layers and another three models using convolutional layers only. The ensemble of IEA of convolutional layers models shows a better performance than the convolutional-layer-only models.

Table 5: Test set mean error rates and standard deviation in percentage on the CIFAR-10 dataset.

| Model | $m = 1$ | $m = 3$ |
|---|---|---|
| VGG16 | 9.88± 0.16 | **9.24± 0.29** |
| ResNet18 | 10.58± 0.14 | **9.58± 0.02** |
| ResNet101 | 9.39± 0.20 | **8.87± 0.34** |
| MobileNet | 13.50± 0.05 | **10.41± 0.23** |
| DenseNet | 7.50± 0.07 | **7.37± 0.10** |
| Wide ResNet | 5.43± 0.63 | **4.47± 0.03** |
| ResNext | 4.29± 0.13 | **3.29± 0.05** |

Table 6: Test set average ensemble error rates in percentage on the CIFAR-10 dataset.

| Model | $m = 1, k = 3$ | $m = 3, k = 3$ |
|---|---|---|
| VGG16 | 8.36 | **7.97** |
| ResNet18 | 9.26 | **8.49** |
| ResNet101 | 7.69 | **7.59** |
| MobileNet | 10.95 | **8.16** |
| DenseNet | 6.31 | **6.01** |
| Wide ResNet | 4.36 | **3.72** |
| ResNext | 3.81 | **3.03** |

### 4.3 CIFAR-100 DATASET

The property images in the CIFAR-100 dataset is identical to that of the CIFAR-10, except that it has 100 classes containing 600 images each. There are 500 training images and 100 testing images

per class. Compared to previously investigated datasets, the CIFAR-100 dataset is much larger and models can truly manifest their power when benchmarked against it. Note that the CIFAR-100 dataset here is augmented using image translation and/or mirroring.

In this section, we plug IEA into Wide ResNet and ResNext and train them on the CIFAR-100. In Table 7, our experiments show that even if our implementation of both networks perform worse than the original paper, our Wide ResNet and ResNext with IEA achieve considerable results. Spcecifically, Wide ResNet with IEA decreases the error rate from 18.85% (Zagoruyko & Komodakis, 2016) to 18.03% on CIFAR-100. ResNext with IEA achieves performance on par with ResNext (Xie et al., 2017). Further, we use outer ensemble to combine individual networks and show in Table 8 that IEA will by no means exhaust the power of variance reduction and outer ensemble can still obtain considerable performance boost. Our results confirm that IEA can still boost the model performance on large scale datasets. The training and validation curves can be found in the supplementary materials.

Also, in order to understand the computational overhead of using IEA table 9 states the inference time per image in milliseconds over the CIFARA-100 validation data set. It can be seen that $m = 3$ Wide ResNet inference time is the same as $m = 1$ ResNext. Meanwhile, we can tell that IEA introduces almost $m$ times computational overhead when used and this is a future research point.

Table 7: Test set mean error rates and standard deviation in percentage on the CIFAR-100 dataset.

| Model | $m = 1$ | $m = 3$ |
|---|---|---|
| Wide ResNet | $22.56 \pm 0.75$ | $\mathbf{18.03 \pm 0.19}$ |
| ResNext | $20.43 \pm 0.33$ | $\mathbf{17.67 \pm 0.33}$ |

Table 8: Test set average ensemble error rates in percentage on the CIFAR-100 dataset.

| Model | $m = 1, k = 3$ | $m = 3, k = 3$ |
|---|---|---|
| Wide ResNet | 19.71 | **15.84** |
| ResNext | 18.28 | **16.44** |

Table 9: Inference time for models trained on CIFAR-100 in milliseconds per image using NVIDIA GeForce GTX 1080Ti GPU.

| Model | $m = 1$ | $m = 3$ |
|---|---|---|
| Wide ResNet | 0.8 ms | 2.36 ms |
| ResNext | 2.1 ms | 6.94 ms |

## 5 VISUALIZATION AND ANALYSIS OF IEA FEATURES

### 5.1 VISUAL INTERPRETATION OF IEA FEATURES

To understand how IEA works and produces better results than ordinary convolutional layer, we visualized the features generated by both. Table 10 shows different features generated by IEA layers and convolutional layer. It is noticeable that the features generated by IEA tend to be more unique and different from each other, unlike convolutional layers where some features appear to be identical. Also, in the first row of Table 10 where the deep model is only 1 layer deep, there exist some IEA features that are associated with zero weight after training. This demonstrates that IEA removed some unnecessary features from the model, which helps improving robustness.

Table 10: Features generated by vanilla CNN with IEA components and regular convolutional layers. Row 1, 2, 3 of the table respectively show the features from the first layer of one, two and three-layered deep models. The models were trained on the rotated-MNIST dataset. The input images are shown in a gray scale while the features are shown using a heat map color. The black shade in the heat map indicates a minimum pixel value, while the white color indicates a maximum pixel value.

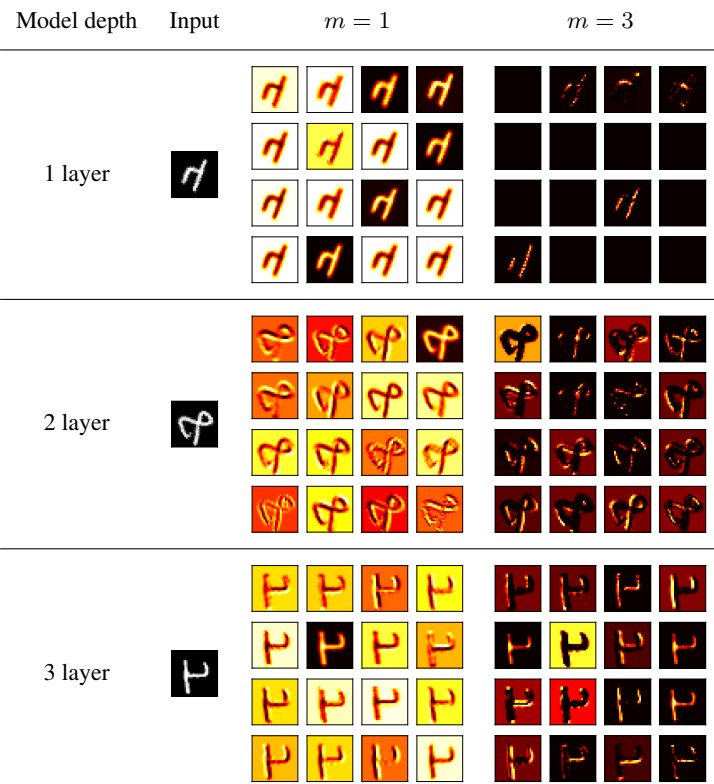

Table 11: The mss-scores of features generated by convolutional layers and IEAs. The score is measured for the first layer only in the models. The models were trained on rotated-MNIST dataset.

| Model depth | $m = 1$ | $m = 3$ |
| --- | --- | --- |
| 1 layer | 0.034 | **2.797** |
| 2 layers | 0.020 | **0.025** |
| 3 layers | 0.020 | **0.038** |

## 5.2 SIMILARITY SCORES

We use similarity scores $\lambda$ to validate the visual interpretation of the IEA features. The similarity score measures how similar an image to another image. If both image of comparison are the same, the similarity score value will be zero. The more difference between the images the more the value of similarity score increases till it reaches the maximum value which is 1. Thus, the usage of similarity score is handful to measure how unique a feature compared to other features produced by the same layer. We introduce the mean sum of similarity score (mss-score) $\mu_\lambda$ as follows: for each $n$ features $f$ in a layer the mss-score is defined as:

$$\mu_\lambda = \frac{1}{n}(\sum_{i=1}^{n} \sum_{j=1, i \neq j}^{n} \lambda(f_i, f_j)) \qquad (2)$$

The higher the mss-score is, the more unique the features produced by the model are. For the similarity score measurement, we used the similarity score model introduced by (Zhang et al., 2018)

Table 12: Features generated by vanilla CNN with IEA components and the same model using MaxOut. The models were trained on the rotated-MNIST dataset. The input images are shown in a gray scale while the features are shown using a heat map color. The black shade in the heat map indicates a minimum pixel value, while the white color indicates a maximum pixel value.

which is proven to outperform any previous similarity score measurements methods. In Table 11, the IEA and convolutional layer mss-scores were evaluated on a batch of 100 validation samples from the rotated-MNIST dataset. The IEAs mss-scores are always greater than the convolutional layer mss-scores. This indicates that the usage of IEA produces more unique features compared to convolutional layer features, and it confirms our visual analysis.

### 5.3 COMPARISON WITH MAXOUT FEATURES

To develop the understanding of the features generated by using IEA, we compare IEA features against the features generated by MaxOut. A one layer vanilla CNN with 48 features channels were trained using MaxOut, the max parameter of MaxOut is set to 3 leading to a network that is similar in the parameters size to the IEA one. By setting the max parameter to 3, the 48 features turns into 16 features, this fortifies a fair comparison with the IEA features on the same model. The training settings were the same as mentioned in section 4.1 but with a lower learning rate set to 0.001 initially.

The trained MaxOut model had an error rate of $21.86\%$ on the rotated-MNIST validation dataset. Visuallay, we can tell that the MaxOut features are more unique than the Vanilla CNN features but they have some similarity in between. Also, MaxOut did not produce such a unique set of features like IEA. One expects that the MaxOut features mss-score to be larger than Vanilla CNN ones, when we computed it had a value of $0.01084$ almost the third of the mss-score value of Vanilla CNN mss-score obtained from table 11. This was surprisingly to us. To investigate this behavior, We state these two facts. Firstly, we calculate the mss-score over a validation batch of 100 images. Secondly, because the MaxOut chooses the max filter between groups this will results in a different model for each input.This suggest that each input image will have a different similarity score as the model behavior changes based on the input.This could lead in total to obtain a low mss-score, also it suggests that it is a drawback of MaxOut compared to IEA.

## 6 CONCLUSION

IEA is a concept that is simple but powerful. It helps the CNN architecture to increase its prediction power by forcing the model to produce more unique features. The cost of using IEA in a CNN is on the model parameter size, yet it is not a significance cost to the improvement in the performance. We showed empirically, visually and by using a similarity scores that the usage of IEA improves the CNN accuracy and produces unique features. Also, the ensemble of IEA of convolutional layers models outperforms the ensemble of convolutional-layer-only model which is a method of inner and outer ensemble. We recommend the usage of IEA where it applies, and a further study of other methods of ensembles shall be conducted.

Our future work also include minimizing the inference time of IEA by finding a criteria for disabling the non-important features and expanding our tests to other not-vision based domains.

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

## Supplementary Materials

# 1 TRAINING AND VALIDATION CURVES

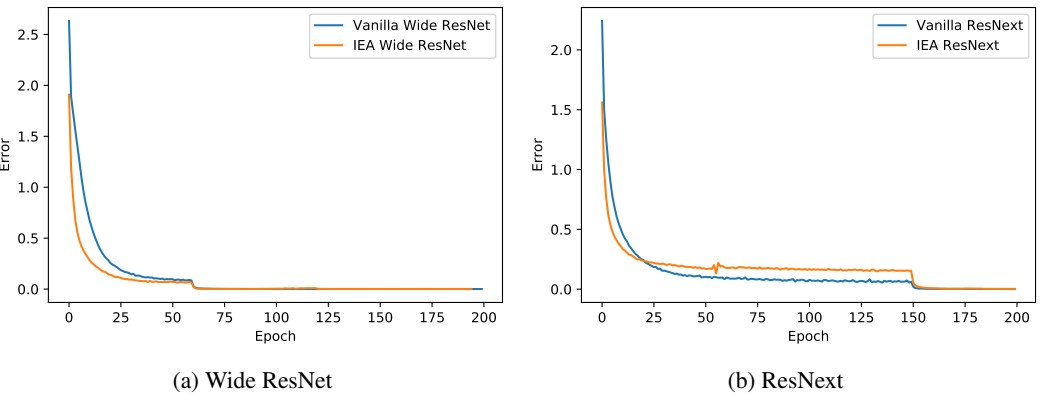

(a) Wide ResNet  (b) ResNext

Figure 1: CIFAR-10 Training curves.

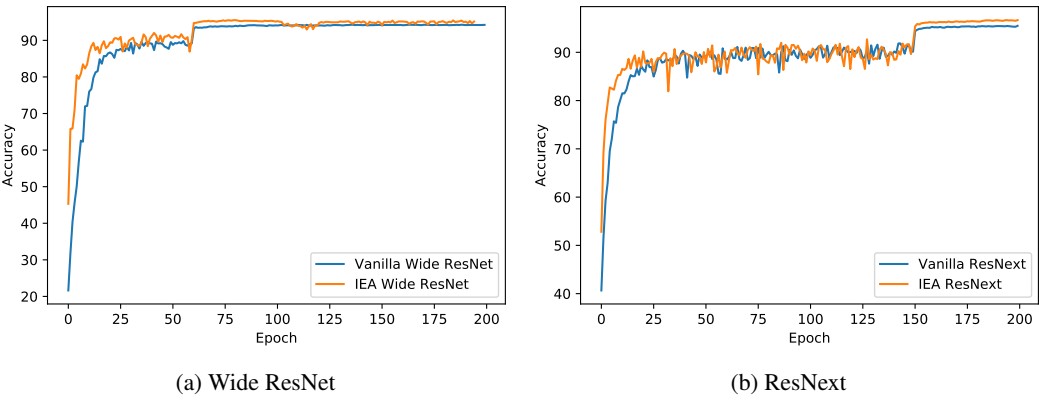

(a) Wide ResNet  (b) ResNext

Figure 2: CIFAR-10 validation curves.

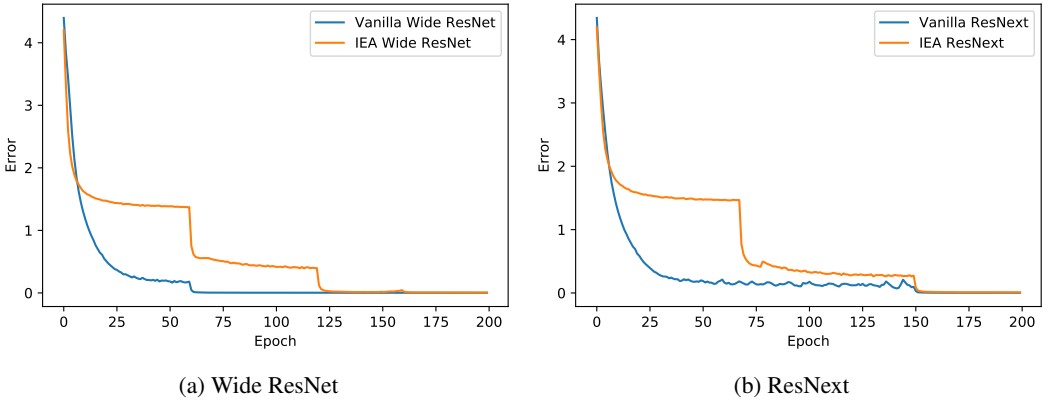

(a) Wide ResNet        (b) ResNext

Figure 3: CIFAR-100 Training curves.

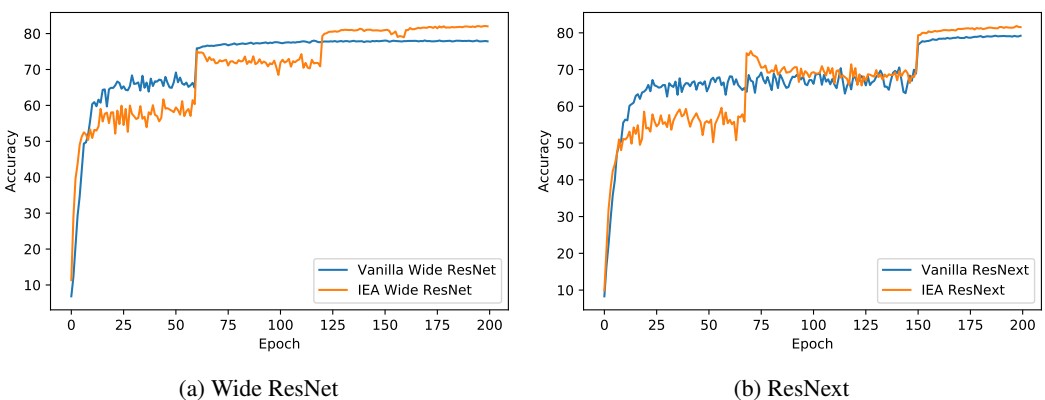

(a) Wide ResNet        (b) ResNext

Figure 4: CIFAR-100 validation curves.

