# OpenReview forum: "IEA: Inner Ensemble Average within a convolutional neural network"
_ICLR.cc/2019/Conference_

### Official Review · AnonReviewer1 · 2018-11-02

**Rating:** 4
**Confidence:** 5

**Review:**

This paper describes a new op: Inner Average Ensemble (IAE). This op is constructed by $m$ convolution ops followed by an averaging op. The author claims using this IAE op is able to improve CNN classification performance. The experiments include MNIST and CIFAR-10 with a few network structures.

First of all, this new proposed op is not efficient. Replace a traditional conv layer with one IAE layer it will introduce $m$ times more parameters, while, the performance gain from the authors’ experiment is relatively small, which indicates, most learning capacity is wasted.

Secondly, only MNIST and CIFAR-10 is not convincing that this structure change will be widely useful.

Thirdly, this work is not practical to apply on real tasks, because it introduced $m - 1$ times more computation.

Overall, I am not convinced this structure change meets innovation standard.

---

> ### Author Response · Authors · 2018-11-26
> **Larger dataset results with a general increase in performance and covering the concerns**
>
> We want to thank the reviewer for his review and dedicating time for our work.
>
> >> Regards first and second point
>
> For the first and second point we updated our work with extra experiments including CIFAR-10 + and CIFAR-100 + on state of the art methods models like WideResNet and ResNeXt. We show that by adapting IEA into these models, WideResNet trained on CIFAR-100+ is able to improve from 18.85 to 18.03 error rate and with ensemble of IEA models we achieve an error rate of 15.84. Also, in ResNeXt IEA  trained on CIFAR-100+ achieveses a performance of 17.44 which on par with the mean of 10 runs in their paper valued at 17.44 error rate and when we ensemble IEAs we obtain an error rate of 16.44. Similar performance weer shown on CIFAR-10. Please note that we also re-trained normal WideResNet and ResNeXt following each work settings and we obtained a less results than what was in the papers. Yet, same configurations were used to train the IEA version of these two models and we achieved the aforementioned results.
>
> >> Regards the third point
>
> We understand your concern regards increasing the parameters size $m-1$ times, we highlighted this in our paper and we follow the same lead of MaxOut and other deep models that expands in depth and width and with the current advancements of GPUs and parallelization, computational time is no longer a bottleneck. Also, we computed the inference time of our model for example WideResNet-IEA inference time on GTX1080Ti is 2.36 ms which is on par with a regular DenseNet model on the same GPU with inference time of 2.09 ms. We also added in our paper the inference time of each model.
>
> >> Conclusion
>
> We hope that our update regards the performance enhancement of our model and the study of inference time accompanied with the unique features generated from the usage of IEA which is shown in the visualization section meets the innovation standard and shows the feasibility of this method.

---

### Official Review · AnonReviewer3 · 2018-11-03
**Issues of clarity and comparison**

**Rating:** 2
**Confidence:** 4

**Review:**

This work proposes an ensemble method for convolutional neural networks wherein each convolutional layer is replicated m times and the resulting activations are averaged layerwise.

There are a few issues that undermine the conclusion that this simple method is an improvement over full-model ensembles:
	1. Equation (1) is unclear on the definition of C_layer, a critical detail. In the context, C_layer could be weights, activations before the nonlinearity/pooling/batch-norm, or activations after the nonlinearity/pooling/batch-norm. Averaging only makes sense after some form of non-linearity, otherwise the “ensemble” is merely a linear operation, so hopefully it’s the latter.
	2. The headings in the results tables could be clarified. To be sure that I am understanding them correctly, I’ll propose a new notation here. Please note in the comments if I’ve misunderstood! Since “m” is used to represent the number of convolutional layer replications, let’s use “k” to represent the number of full model replications. So, instead of “CNL” and “IEA (ours)” in Table 1 and “Ensemble of models using CNL” and “Ensemble of models using IEA (ours)” in Table 2, I would recommend a single table with these headings: “(m=1, k=1)”,  “(m=3, k=1)”,  “(m=1, k=3)”,  and “(m=3, k=3)”, corresponding to the columns in Tables 1 and 2 in order. Likewise for Tables 3-6.
	3. Under this interpretation of the tables---again, correct me if I’m wrong---the proper comparison would be “IEA (ours)” versus “Ensemble of models using CNL”, or  “(m=3, k=1)” versus “(m=1, k=3)” in my notation. This pair share a similar amount of computation and a similar number of parameters. (The k=3 model would be slightly larger on account of any fully-connected layers.) In this case, the “outer ensemble” wins handily in 4 of 5 cases for CIFAR-10.
	4. The CNL results, or “(k=1,m=1)”, seem to not be state-of-the-art, adding more uncertainty to the evaluation. See, for instance, https://www.github.com/kuangliu/pytorch-cifar. Apologies that this isn’t a published table. A quick scan of the DenseNets paper and another didn’t yield a matching set of models. In any case, the lack of data augmentation may account for this disparity, but can easily be remedied.

Given the above issues of clarity and that this simple method seems to not make a favorable comparison to the comparable ensemble baseline (significance), I can’t recommend acceptance at this time.

Other notes:
	* The wrong LaTeX citation function is used, yielding the “author (year)” form (produced by \citet), instead of “(author, year)” (produced by \citep), which seems to be intended. It’s possible that \cite defaults to \citet.
	* The acronyms CNL and FCL hurt the readability a bit. Since there is ample space available, spelling out “convolutional layer” and “fully-connected layer” would be preferred.
	* Other additions to the evaluation could or should include: a plot of test error vs. number of parameters/FLOPS/inference time; additional challenging datasets including CIFAR-100, SVHN, and ImageNet; and consideration of other ways to use additional parameters or computation, such as increased depth or width (perhaps the various depths of ResNet would be useful here).

---

> ### Author Response · Authors · 2018-11-26
> **Exceeding SOTA, larger datasest tests and general improvements**
>
> We want to thank you for putting such an effort in providing us with this review.
>
> >>Regards clarity of the concept
>
> Thank you very much for this remark, in our paper  IEA is indeed applied after a nonlinear operation,  in style IEA itself is a plug and play method inside deep models.
>
> >> Regards adapting the m and k for enhancing the readability
>
> Thanks for mentioning this, we adapted this approach in the updated version of the paper.
>
> >> Regards the comparison with ordinary ensemble
>
> Our goal is not to state that the Inner ensemble is better than outer ensemble. Our main goal is to introduce this operator and show its ability to boost the performance wherever it used.
>
> >> Regards SOTA and expanding the tests to larger datasets
>
> We used the rebuttal period to test IEA on CIFAR-100+ and CIFAR-10+. Our results on CIFAR-100+ show that using IEA inside WideResNet improves  the error rate from 18.85 according to  WideResNet paper to 18.03. Also, for WideResNet when we have the case of (m=3,k=3) we obtain an error rate of  15.84.  We also tested IEA on ResNeXt and we had an error rate of 17.44  which is on the par with the error rate reported in their paper of 17.31. In the case of (m=3,k=3) for ResNeXt, we obtain an error rate of 16.44 which also exceeds their work. Please note that we also re-trained normal WideResNet and ResNeXt following each paper training settings and we obtained a results that is less than what was reported in their the papers. Yet, by using the same configurations in their papers the IEA version of these WideResNet and ResNext achieved the aforementioned results. We also achieved similar results on CIFAR-10+.
>
> >>Conclusion
>
>
> We hope that our new results which shows a performance increase by using IEA and the addition of inference measurements, editing the paper structure to increase clarity matches your expectations for what this work should look like.
>
> >> Other notes
> Other notes:
>
>
> We fixed \citep  problem.
>
> We changed the FCL acronym this, it’s much better, thanks again for suggesting this.
>
> We added the results of training on CIFAR-10+ and CIFAR-100+, for ImageNet we couldn’t train ourmodels due to lack of resources and time. We also added training, testing error curves and inference time.
>
> Thanks again

---

> > ### Comment · AnonReviewer3 · 2018-12-10
> > **Response to Updates**
> >
> > Thanks for your response and the updated experiments and paper text, especially the experiments with augmentation on CIFAR.
> >
> > My primary concern is the lack of comparison between your layer-wise ensembles and full-network ensembles, e.g. “(m=3, k=1)” versus “(m=1, k=3)”. This issue is coupled with the lack of a comparison to maxout networks, as pointed out by another reviewer. Given these, I unfortunately can't raise my score.

---

### Official Review · AnonReviewer4 · 2018-11-08
**A number of missing comparisons, needs stronger empirical results**

**Rating:** 4
**Confidence:** 3

**Review:**

IEA proposes to use multiple "parallel" convolution groups, which are then averaged to improve performance.

This fundamental idea of ensembles combined with simple functions has been explored in detail in Maxout (Goodfellow et.  al., https://arxiv.org/abs/1302.4389) in the context of learning activation functions, and greater integration with dropout regularization.

Under the lens of comparison to Maxout (which should be cited, and is a key comparison point for this work), a number of questions emerge. Does IEA also work for feedforward layers? Does IEA give any performance improvement or have some fundamental synergy with the regularizers used here? Is the performance boost greater than simply using an ensemble of m networks directly (resulting in the equivalent number of parameters overall)? The choice of the mean here seems insufficient for creating the types of complexity in activation which are normally desirable for neural networks, so some description of why a simple mean is a good choice would be beneficial since many, many other functions are possible.

Crucially Maxout seems much too close to this work, and I would like to see an indepth comparison (since it appears to be use of mean() instead of max() is the primary difference). I would also significantly reduce the claims of novelty, such as "We introduce the usage of such methods, specifically ensemble average inside Convolutional Neural Networks (CNNs) architectures." in the abstract, given that this is the exact idea explored in other work including followups to Maxout.

For example, MNIST performance here matches Maxout (.45% for both, but Maxout uses techniques known in 2013). CIFAR-10 results are better, but again Maxout first appeared 5 years ago. There are more recent followups that continued on the line of work first shown in Maxout, and there should be some greater comparison and literature review on these papers. The CIFAR-10 baseline numbers are not ideal, and since IEA is basically "plug and play" in existing architectures, starting from one of these settings instead (such as Wide ResNet https://arxiv.org/abs/1605.07146) and showing a boost would be a stronger indication that this method actually improves results. In addition, there are a number of non-image settings where CNNs are used (text or audio), and showing this idea works on multiple domains would also be good.

There seems to be a similarity between ResNet and this method - specifically assuming the residual pathway is convolution with an identity activation, the summation that combines the two pathways bears a similarity to IEA. With multiple combined paths (as in Dense ResNet) this equivalence seems stronger still. A discussion of this comparison in greater detail, or even derivation of IEA as a special setting or extension of ResNet (coupled with stronger performance on the datasets) would help ground the work in prior publication.

The section on visualization and inspection of IEA features seems interesting, but too brief. A greater exploration of this, and possible reduction or removal of the ensemble selection section (which didn't have a clear contribution to the message of the paper, in my opinion) would strengthen the work - and again, comparisons to activations learned by Maxout and followups would make this inspection much stronger.

My key concerns here are on relation to past work, greater comparison to closely related methods, and improvement of baselines results. Given the close similarity of this work to Maxout and others, a much stronger indication of the benefits and improvements of IEA seems necessary to prove out the concepts here.

---

> ### Author Response · Authors · 2018-11-26
> **Covering the concerns, updated results and raising SOTA bar**
>
>
>
> Dear reviewer,
>
> Honestly, We thank you for such a detailed review and pointing out such an important work.
>
> >>Regards the first paragraph of your review
>
> By using IEA combined with augmentation and dropout we were able to bypass current SOTA on CIFAR100 and we show in the updated version of the paper these results on both WideResNet and ResNeXt. We also show that the IEA in some cases bypass an ensemble of m networks. To clear something, we don’t directly compare IEA with ensemble rather than introducing a new operator that can enhance the current models wherever it applied. We also show that the ensemble of IEAs boost these models results more.
>
> Our intuition for choosing mean is as follow: Average ensemble of models are used essentially to reduce the variance, in other terms to increase models generalization ability. We hypothesize that the overall variance of the model can be decomposed into sub-layers within the model and each layer contributes somehow into this variance. If we used inner ensemble average we will reduce the sub-variances of each layer resulting in an overall variance reduction per model. To support our hypothesis is correct we show that features generated by IEA are unique and it suppresses the features that causes the model ambiguity or variance and this is discussed in the visualization section. For feedforward layers, we didn’t test IEA on it as our main focus is towards convolutional layers.
>
> >>Regards Maxout
>
> “Max out in essence choose a winner unit that will receive the gradients and other units will have an error of zero and in corner case when there is a tie for winner, the function is not differentiable, because the left-sided derivative does not match the right-sided derivative for each of the elements that ties” [1]. This will require some special implementation to solve this issue also it will leave the network with ambiguous filters that will increase the overall variance because they weren’t trained on all of the training set. Unlike in our approach, were we use the average, the gradients are distributed equally amongst the filters and this what leads to unique features as shown in the visualization section. We also changed the abstract following your directions.
>
> >>Regards related work
>
> We added a literature review following your directions.
>
> >>Regards results
>
> We added results in our updated version that shows performance boost on WideResNet and ResNeXt too. For other domains like text and audio we believe that IEA will also boost the performance following the same leads on vision task, still we strongly agree that having results on non-image settings will be a positive thing for the work, yet we currently don’t have the means of doing such experiments.
>
>
> >>Regards similarity with ResNet and DesnsNet
>
>
> In the literature review we discussed this. But IEA in essence as you mentioned is a plug and play. For example in ResNet the activation function is $H(x) = x+ F(x)$ and for DenseNet we can write it as $F_i(x) = F_{i-1}(F_{i-2}(...F_0(x))) $. IEA itself cares about F(x) and treats it as average of multiple convolutional neural layer, so IEA is an addon to these architectures more than something to compare against. Also, our latest results supports the idea of IEA.
>
> >>Regards the visualization section
>
> We removed the ensemble selection section as we agree on this.  The visualization section is concerned with comparison of IEA versus ordinary CNN. We also investigated more by using the similarity score introduced in the paper. Our main aim of this section is to have an intuition behind IEA and why it reduces the model variances. We added a comparison between the MaxOut features and IEA features and we had some interesting findings in the updated version of the paper, thanks again for suggesting this.
>
> >>Conclusion
>
> We hope that our updates matches your expectations and your concerns were covered, also we would like to thank you a lot for such a feedback!
>
>
>
> [1]https://www.reddit.com/r/MachineLearning/comments/2vl8hp/a_question_about_maxout_who_gets_the_error/coj1o5y/

---

> > ### Comment · AnonReviewer4 · 2018-11-28
> > **Thank you for your updates**
> >
> > The current experiments are much improved, thank you for doing the work required in such a limited timeframe. Regarding the hypothesis, it would be good to directly design a specific toy experiment to test this idea, as it would lead to better intuition about why it helps as well as potentially guiding followup work. As for as SOTA discussion, I would be careful making that claim as ShakeDrop regularization seems to have a far better score in the CIFAR-100 case. SOTA is definitely not a necessary claim for a paper or method in my opinion, but if you wish to make that claim be sure it is true. In a further iteration of the paper I would suggest combining some of the result tables together, as the current results seem extremely space inefficient, disrupting the flow and format of the work.
> >
> > I don't see the difficulty in implementing Maxout here as nearly all frameworks can directly implement Maxout as a composition of linear ops with max, and without a comparison directly to Maxout it is difficult to say whether the improvements over baselines are due to other effects - Maxout is a simple, direct comparison which has so much in common with IEA (with regards to parameter count, computational complexity, etc.) it could really be useful. See for example the discussion of implementation here (https://groups.google.com/forum/#!topic/caffe-users/KOttrPG3vUA), it is generally easier in other frameworks than caffe and most have direct implementations to use.
> >
> > The added results are good, though the deviation in performance from the reference implementations online (for example, Wide ResNet) is still a bit troubling since the IEA "boost" is still below the reported and replicated results for some models. If you are reimplementing these other models directly, it may be easier to implement IEA within the existing codebases to test quickly - this would further isolate the performance of IEA to one change on existing, known code and results.
> >
> > Overall, the newer version is much improved compared to the older work but without the chief concern being directly addressed (comparison directly with Maxout, and direct plug-in to a known replicated baseline) I cannot improve my score. However I commend the authors on their hard work improving the paper - it definitely shows!

---

### Author Response · Authors · 2018-11-26
**Updated version**

We updated a 3rd version correcting some grammatical errors.

---

### Meta-Review · Area_Chair1 · 2018-12-13
**Interesting method with limited novelty and requiring better baselines.**

**Confidence:** 4
**Recommendation:** Reject

**Metareview:**

The method under consideration uses parallel convolutional filter groups per layer, where activations are averaged between the groups, forming "inner ensembles".

Reviewers raised a number of concerns, including the increased computational cost for apparently little performance gain, the choice of base architecture (later addressed with additional experiments using WideResNet and ResNeXt), issues of clarity of presentation (some of which were addressed). One reviewer was unconvinced without direct comparison to full ensembles. Another reviewer raised the issue of a missing direct comparison to the most similar method in the literature, maxout (Goodfellow et al, 2013). Authors rebutted this by claiming that maxout is difficult to implement and offering vague arguments for its inferiority to their method.

The AC agrees that a maxout baseline is important here, as it is extremely close to the proposed method and also trivially implemented, and that in light of maxout (and other related methods) the degree of novelty is limited.  The AC also concurs that a full ensemble baseline would strengthen the paper's claims. In the absence of either of these the AC concurs with the reviewers that this work is not suitable for publication at this time.